# How Do Mixed Cover Crops (White Mustard + Oats) Contribute to Labile Carbon Pools in an Organic Cropping System in Serbia?

**DOI:** 10.3390/plants13071020

**Published:** 2024-04-03

**Authors:** Vladan Ugrenović, Vladimir Filipović, Vladimir Miladinović, Divna Simić, Snežana Janković, Slađan Stanković, Elmira Saljnikov

**Affiliations:** 1Department of Soil Amelioration, Institute of Soil Science, Teodora Drajzera 7, 11000 Belgrade, Serbia; vladan.ugrenovic@gmail.com (V.U.); vladimir.miladinovic33@gmail.com (V.M.); 2Institute for Medicinal Plant Research “Dr. Josif Pančić”, T. Košćuška 1, 11000 Belgrade, Serbia; vfilipovic@mocbilja.rs; 3Institute for Science Application in Agriculture, 68b Blvd Despota Stefana, 11000 Belgrade, Serbia; simic.divna@yahoo.com (D.S.); sjankovic@ipn.bg.ac.rs (S.J.); sstankovic@ipn.bg.ac.rs (S.S.); 4Mitscherlich Academy for Soil Fertility (MITAK), Prof.-Mitscherlich-Allee 1, 14641 Paulinenaue, Germany

**Keywords:** organic farming, soil respiration, microorganisms

## Abstract

Sustainable farming is one of the priority goals of the “4 per 1000” concept with regard to the preservation of soil fertility and carbon sequestration. This paper presents a study on the use of a mixture of cover crops of self-grown oats (*Avena sativa* L.) and sown white mustard (*Sinapis alba* L.) in organic farming under the agroecological conditions of Serbia. The main objective was to identify sensitive carbon pools (microbial carbon and nitrogen, basal respiration and a number of specific groups of soil microorganisms) in organic farming with and without cover crops. The inclusion of a mixture of white mustard and self-grown oats as a cover crop led to a significantly increased biogenity of the soil compared to a control after only a few years of investigation. The number of microorganisms, soil respiration and microbial biomass carbon were significantly higher in the cover crop treatment compared to the control soil on an organic farm in Serbia. This is the first study in Serbia to investigate the effect of self-grown oats as a cover crop. Further research will incorporate a wider range of variables and factors in order to develop a sustainable and effective site-specific system for organic crop production in Serbia.

## 1. Introduction

Current trends in soil use and climate change have led to a loss of soil organic carbon (OC) at a rate equivalent to 10% of total fossil fuel emissions for Europe as a whole [1,2]. Soil organic carbon (SOC) sequestration in agricultural soil is directly affected by anthropogenic activities and climate change; both can alter net primary production and organic matter decomposition. The results of many studies indicate that these processes are largely influenced by agronomic practices: conventional tillage (plowing), intensification of production (narrow crop rotations and application of synthetic mineral fertilizers), improper management of crop residues (carrying out and burning), etc., [3,4,5,6,7,8,9,10].

Organic farming is a powerful means to combat organic carbon loss from soil and at the same time to contribute to carbon sequestration and mitigation of CO_2_ emissions from soil to air. The European Commission adopted “A Farm to Fork Strategy” COM (2020) 381 [11] and “EU Biodiversity for 2030” COM (2020) 380 [12], which comprehensively respond to the challenges of sustainable food systems. In Serbia, organic farming has been practiced since the early 1990s [13] and has developed further since then. Studies in Serbia have shown that the use of cover crops has many positive effects on soil and plant quality as well as on the environment [14,15,16,17,18,19]. However, there are still many questions that need to be investigated and verified for the region. This is important because the natural conditions in Serbia offer great potential for the development of organic crop production.

Considering that the activity and biomass of soil microorganisms are the most sensitive indicators of changes in the environment and are key parameters controlling the short-term carbon cycle in the terrestrial ecosystem [20], the assessment of parameters such as basal respiration, microbial carbon and nitrogen biomass, microbial metabolic quotient and specific groups of microorganisms is of great importance for monitoring subtle changes in soil quality.

The quality and quantity of the labile fraction of SOM are highly dependent on seasonal changes in plant and crop residues. Soil carbon monitoring results suggest that targets for potential carbon sequestration through soil management practices need to be refined [21,22], including expanding the supply of regionally specific crops, including cover crops. Since the use of synthetic fertilizers is strictly limited in organic farming, it is important to increase the natural fertility of the soil through the activities of soil organisms (bacteria, fungi, etc.). In this sense, cover crops grown all year round allow more carbon to be sequestered in the soil, thus reducing the amount of CO_2_ released from the soil. Although organic farming has been practiced in Serbia for decades, there is still a lack of research on its effects on labile soil carbon stocks in relation to crop productivity, soil quality and climate change.

The main research goal was to study self-grown oats (*Avena sativa* L.) in combination with white mustard (*Sinapis alba* L.) as a cover crop, which is returned to the soil as green biomass. The objectives were: 1. to observe the development of the cover crops, their productivity and quality; 2. to evaluate the distribution of sensitive carbon pools in organic farming with and without cover crops.

## 2. Results

### 2.1. Soil Properties

The main agrochemical soil parameters at the start of the experiment in 2016 and at the time of sampling in 2020 are shown in Table 1. As expected, soil agrochemical parameters did not change after the 4 years of the experiment. However, there was a clear trend towards greater soil organic carbon and nitrogen accumulation in the OF-CC cover crop treatments.

### 2.2. Land Cover Dynamics, Biomass Productivity and Quality

After shallow processing of the stubble on 1 July 2017, self-grown oats emerged on 10 July (BBCH 09). This was followed by the sowing of white mustard on 15 July, which emerged on 3 August (BBCH 09). In 2019, after shallow tillage of the stubble on 17 July, the self-growth oats emerged on 25 July. White mustard, on the other hand, was sown on 2 August and emerged on 7 August (BBCH 10). In both years, therefore, a cover crop of self-grown oats and sown white mustard was established. The oat plants grew as individual shoots until stage 20 BBCH, did not go through any further stages, and remained green until the cover crop was destroyed for mulching. It was found that the self-grown oats contributed to soil coverage as an inter-crop, especially in the early stages of the formation of mustard cover crop. The rapid growth of white mustard contributed to the closure of the rows, which occurred on 20 August 2017 and on 27 August 2019, respectively (Figure 1).

At this time, the measured ground coverage was 80%. The ground coverage increased over time and reached 100% at the elongation stage of white mustard (BBCH 34). The white mustard began flowering (BBCH 60) on 20 October 2017 and on 25 October 2019. The main tillage coincided with the flowering stage of 30% of the white mustard plants (BBCH 63) on 20 November 2017 and 25 November 2019. This is very important because, according to [23], the biomass of the flowering plants is rich in easily soluble compounds that decompose quickly in the soil due to the narrower C:N ratio. The dynamics of cover crop biomass formation with self-grown oats and sown white mustard are shown in Figure 2.

Cover crop biomass was destroyed for mulching with a roller (IQ Storm M 3000 roller, IQ Patent, Subotica, Serbia) on 20 November 2017 and 25 November 2019, followed by deep tillage. The average total yield of cover crop biomass at the time of destruction was 9.8 t ha^−1^. The average biomass yield of oats was 1.5 t ha^−1^ and of white mustard was 8.3 t ha^−1^. The total yield of green biomass was 9.0 t ha^−1^ in 2017 and was 10.61 t ha^−1^ in 2019 (Table 2).

### 2.3. Microbial Carbon and Soil Respiration

Soil respiration rates were significantly higher in the treatments with cover crops (average 1065.8 μg/g CO_2_-C/week) compared to the control soil without cover crops (average 469.33 μg/g CO_2_-C/week). The microbial biomass carbon (MBC) was also significantly higher in the treatments with cover crops (235.28 μg/g) compared to the control without cover crops (159.83 μg/g). However, the nitrogen content in the microbial biomass (MBN) did not differ significantly. The MBC/MBN ratio was 1.25 times higher in the cover crop treatment than in the control (Table 3). The amount of microbial C in the biomass and the amount of mineralized C in the treatment with cover crops was on average 1.47 and 1.54 times higher than in the control, respectively.

Soil respiration was positively correlated with MBC, MBN, carbonates, organic carbon and total nitrogen (Table 4). MBC was positively correlated with organic carbon and total nitrogen as well as with carbonates and pH. MBN was highly correlated with the same parameters as MBC, with the exception of total nitrogen.

The amount of carbon returned to the soil by cover crop biomass was 788.8 t ha^−1^ (Table 5). Mineralization of OM by microbial respiration was 1.935 t ha^−1^ week^−1^, and microbial tissues incorporated 0.427 t ha^−1^ C.

Cover crop cultivation significantly increased the total number of microorganisms (35.21 × 10^6^ g^−1^) compared to the control soil (10.33 × 10^6^ g^−1^). These results are consistent with the results of respiration intensity (Table 6). The number of ammonifiers increased significantly in the cover crop soils (116.25 × 10^5^ g^−1^) compared to the control (31.50 × 105 g^−1^). This indicates intensive processes of decomposition of organic nitrogen compounds to ammonia nitrogen [4].

The role of the cover crop in improving soil fertility is also evidenced by the significantly higher number of oligonitrophils in the soil with cover crops, which had 95.59 × 10^5^ g^−1^ compared to 49.01 × 10^5^ g^−1^ in the control soil. The number of actinomycetes was also significantly higher in the soil with cover crops (32.58 × 10^4^ g^−1^) than in the control (11.34 × 10^4^ g^−1^).

## 3. Discussion

In general, total soil organic matter and soil acidity do not change significantly over a short period of time. However, the results show that the annual introduction of green biomass from cover crops into the soil can improve some soil parameters after only four years. In contrast to total soil organic matter, labile organic carbon pools respond quickly and at a subtler level to changes in soil management.

According to [23], plant biomass during the flowering phase contains larger amounts of easily soluble compounds than during other phonological phases. These compounds have a narrower C:N ratio and decompose more quickly. Therefore, it is important that the main tillage coincides with the flowering phase of 30% of the white mustard plants (BBCH 63). In general, the biomass of cover crops with a narrower C:N ratio decomposes faster, depending on the plant species and the time since the cover crop was mulched [24]. White mustard and oat plants were mulched and the biomass remained green. The average C:N ratio of the biomass was 8.68:1 for white mustard, 13.78:1 for oats and 9.8:1 for total biomass (Table 2). The lower C:N ratio in the total biomass of both crops was advantageous due to the faster supply of mineral nitrogen to the following crop.

According to the USDA classification, the intensity of soil microbial activity in the soil after the return of cover crop biomass corresponds to the ideal soil activity class, which is between 1000 and 2000 mg/kg soil/week [25]. This indicates that the soil was well supplied with organic matter and had an active microbial population, while the control soil had significantly lower microbial activity corresponding to moderately low soil activity (300–500 mg/kg soil/week). This is confirmed by the correlation coefficients of microbial biomass C and N and respiration indices with the main agrochemical parameters (Table 4). The microbial utilization efficiency of soil carbon, measured as the metabolic quotient (*q*CO_2_), was significantly higher in OF + CC.

The strong correlation of MBN with the same parameters as MBC, but with total nitrogen, is due to the fact that MBN is more dependent on the recent supply of fresh organic substrate than on total organic carbon [26]. Similarly, [27] found that microbial respiration responded linearly to an increased supply of freshly added organic matter. Microorganisms respond rapidly to added crop residues to obtain readily available nitrogen as a nutrient and carbon as an energy source. At the same time, the carbon incorporated into the microbial tissue (MBC) is very sensitive to total carbon and nitrogen stocks [7,20]. Considering the results of OM mineralization by microbial respiration and the amount of carbon incorporated into microbial tissues, and taking into account that the reflux of total cover crop biomass continues, a significant contribution of cover crops to carbon content can be expected.

The higher amount of CO_2_ respired from the fields is directly related to a higher mineralization rate of the added fresh plant biomass. This freshly added organic matter serves as a readily available source of N and C for microorganisms and accelerates their growth and thus microbial respiration [26]. This is confirmed by a significantly higher number of soil microorganisms in the cover crop treatments (Table 6). The results obtained indicate that the addition of white mustard and oats as a cover crop mixture has a very positive effect on the abundance and diversity of soil microorganisms due to the extended crop rotation and the return of green biomass to the soil.

Since soil microorganisms have the narrowest C:N ratio, they are the best available source of nutrients and energy for growing microorganisms after death [28]. Under optimal laboratory conditions, soil microorganisms had a greater source of labile organic compounds as green biomass and microbial necromass in the organic plots. This is supported by higher C mineralization rates. A higher MBC:MBN ratio indicates a higher efficiency of CO_2_ utilization by soil microorganisms [29].

In the long term, traditional crop production is not self-sustaining and often leads to a decline in the OM content of arable soils. When the soil is covered with plants or a mulch layer, it is less susceptible to extreme climatic conditions, water and wind erosion, and loss of soil moisture; the topsoil is protected from the negative effects of high temperatures; the soil aggregate stability is improved [30,31]; and soil biodiversity and organic carbon are preserved [32].

In addition, cover crops help to store soil carbon in deeper soil layers with roots. Mixtures of different plants can penetrate different soil layers better than single plants due to a higher root density and different root vectors. The potential for soil carbon enrichment with cover crops is estimated at 100 to 460 kg C ha^−1^ year^−1^ in the topsoil and 10 to 320 kg C ha^−1^ year^−1^ in the subsoil when cover crops are regularly introduced over a long period of 12 to 50 years [33].

There are still many knowledge gaps regarding the cumulative effect of cover crops on carbon sequestration and their impact on soil and crop yield parameters [31,34]. Many studies have shown that mulching with crop residues increases weed infestation [35], which is particularly acute in organic farming where pesticides are not used. Although this aspect has not yet been fully researched, there are some positive results showing that proper mulching [36], mulching depth and crop diversification [37] can significantly solve this problem.

## 4. Materials and Methods

### 4.1. Experimental Design

The research was carried out in the period from 2016 to 2020 on a certified organic field of the Tamis Institute (44°56′35.3″ N 20°43′08.8″ E), on an area of 1.25 ha on carbonate chernozem, Mollisol [38]. An organic cropping system with a cover crop (a mixture of sown white mustard and self-grown oats—OF + CC) was compared with a control (OF) without cover crops, but with the same crop rotation and agrotechnical techniques. The main crops were winter oats (*Avena sativa* L.), variety NS Jadar, and soybeans (*Glicine max* L.), variety Galeb. No fertilizers or chemicals for pest and weed control were used in the organic fields studied. The trials were set up in a random block system in six replicates; the area of the base plot was 15 m^2^, i.e., 6 m long and 2.5 m wide. A scheme of the crop rotation and experimental plots is shown in Figure 3B.

At the start, basic agrochemical analyses of the soil were carried out in six replications. At the end of the study, in spring 2020, soil samples were taken from the top 10 cm layer in each plot, both for the treatment and for the control. Biological soil indicators that were analyzed were soil respiration, the amount of microbial nitrogen and carbon (MBN, MBC), and the number of groups of microorganisms.

As the main soil tillage, conservation tillage was used with a combined implement (chisel + disc harrow + roller—Horsch Terrano 3 FX, HORSCH Maschinen GmbH, Schwandorf, Germany) to a depth of 10 cm, immediately after harvesting the oats in the first ten days of July. The entire organic field (Figure 3A) was sown with white mustard at the beginning of August using a direct sowing seeder, with a seed rate of 15 kg seed ha^−1^ of the Brizant variety at a row spacing of 12.5 cm at a depth of 2–3 cm. Stationary test plots were then formed on this area (Figure 3B). Control plots without cover crops and weeds were continuously maintained using mechanical methods for the duration of the trial. Cover crop biomass was mulched immediately prior to the fall with deep tillage using an IQ Storm M 3000 (IQ Patent, Subotica, Serbia) roller, and the soil was tilled to a depth of 25 cm using the Horsch Terrano 3 FX implement. Phenological observations using the biological development stage scale (BDSD) were carried out on established cover crops (oats, white mustard) during the growing season in all study years [39]. The grid partitioning method [40] was used to evaluate the soil cover with the oat–white mustard combination. With this method, the shape of the base area (1 × 1 m = 1 m^2^) and the size of the individual squares (10 × 10 cm) were changed. The biomass of the two plant species was also measured immediately before mulching the cover crops.

### 4.2. Climatic Conditions

Data from the Meteorological Institute of Serbia were used to analyze the amount and distribution of precipitation, as well as the thermal regime during the monitoring experiment for the Pancevo location in both study years (Figure 4).

The sum of positive temperatures during the growing season of the cover crops (July–November) was approximately the same in the first (83.4 °C) and the second year of research (87.3 °C). In 2017, the average monthly July temperatures were warm (23.9 °C). Precipitation at the beginning of the cover crop growing season was favorable in both study years. Average precipitation during the cover crop growing season was higher in the second year of study (207.1 mm) compared to the first (160.1 mm). The monthly water regime in the second year was more favorable, with sufficient water during periods of maximum consumption, while in the first year there was a deficit (9.7 mm) in July (Figure 4). In the period after destruction of the biomass of cover crops, when mineralization processes in the soil take place (December–May), the sum of positive temperatures in 2017/18 was 49.2 °C and in 2019/20 it was 45.5 °C. The average amounts of precipitation for this period were 268 mm and 238.2 mm, respectively.

### 4.3. Analytical Methods

Soil sampling was carried out at the beginning and at the end of the study in 2016 and 2019, respectively. Soil total carbon (TC) and nitrogen (TN) were measured on a CNS elemental analyzer (Model Vario EL III-ELEMENTAL Analysis systems, GmbH, Hanau, Germany) by dry combustion at 1150 °C. Soil pH was determined using a pH meter with a glass electrode in 1 mol L^−1^ KCl at a ratio of 1:2.5 (*w*/*v*) and in distilled water with a ratio of 1:20 (*w*/*v*). Phytoavailable potassium (K_2_O) and phosphorus (P_2_O_5_) were determined as described by [41]. Cover crop biomass was sampled immediately prior to mulching to determine total nitrogen and carbon as well as dry weight of biomass.

#### 4.3.1. Basal Soil Respiration

Soil respiration was measured as CO_2_ produced by the mineralization of organic C under controlled laboratory conditions (temperature 28 °C and soil moisture 50% WHC). It should be noted that this parameter is not representative of in situ conditions, but simulates optimal and equal conditions for both treatments studied. The water content of 20 g of fresh soil samples was measured and a uniform moisture content of 50% of the water holding capacity was adjusted for each individual sample. The released carbon dioxide was bound by 0.2 N NaOH, and the amount of residual free NaOH was determined by titration with 0.02 N HCl. The amount of carbon dioxide released for each incubation time was calculated from the difference between the amount of NaOH removed for the bound carbon dioxide and the amount determined by titration with HCl.

#### 4.3.2. Microbial Biomass Carbon and Nitrogen (MBC and MBN)

The carbon and nitrogen of the microbial biomass were determined using the fumigation–incubation method [42]. Soil samples were taken from each field treatment in four replicates. In the laboratory, soil moisture was adjusted to 50% WHC, followed by a 17 h fumigation of 17 g of soil with chloroform in a vacuum desiccator. After the defumigation, 3 g of fresh soil from the same treatment was added as an inoculant. At the same time, 20 g of non-fumigated soil was prepared as a control for a further 5-day incubation under controlled humidity and temperature conditions (28 °C). After incubation, the ammonia and nitrate nitrogen content was extracted from the fumigated and non-fumigated soil samples using 2 M KCl and then determined using the Kjeldahl digestion method. The difference between fumigated and non-fumigated soil was used to calculate the amount of nitrogen and carbon of the living part of the SOM. The microbial metabolic quotient (*q*CO_2_) was calculated by dividing the soil respiration by the MBC.

#### 4.3.3. Specific Groups of Microorganisms

The number of total microflora and fungi in the soil was determined by the indirect method of agar plates, based on the principle of seeding appropriate selective nutrient media by decimal dilutions of the test soil suspension. It should be noted that not all microorganisms are able to grow on agar medium. However, we used this method to comparatively characterize the two experiments studied. Soil samples were preliminarily prepared and sifted through a flame sieve with a diameter of 2–3 mm, and their absolute moisture content was determined. The number of total microflora was determined on agar with soil extract and also the number of fungi on Czapek agar, the number of *Azotobacter* in liquid nitrogen-free mannitol medium (Tchan’s method) and the number of ammonifiers in liquid medium with asparagine [43]. To determine the number of actinomycetes in the soil, a medium with sucrose according to Krasiljnikov was used; free ammonifiers were determined on a medium according to Fyodorov [44]. After incubation of microorganisms for 5–7 days (except for fungi for 3–5 days) at 28 °C, their number was determined and the average number per gram of dry soil was calculated.

### 4.4. Statistical Analysis

One-way ANOVA was followed by a Duncan’s post-hoc test, which was carried out to reveal the significance of the effects of treatment with cover crops on the following soil characteristics. All analyses were carried out with the software SAS/STAT^®^ 13.1, SAS Institute Inc. (Cary, NC, USA) 2014.

## 5. Conclusions

The study showed that the increased application of fresh residues of a mixture of white mustard and oats as a cover crop to the soil significantly increases the number of soil microorganisms, their activity and microbial biomass carbon. This contributes to an increase in crop production, which means that more organic substrate is returned to the soil. A cropping system based on cover crops contributes to achieving carbon sequestration targets, on the one hand, and sustainable soil management, on the other. This study was the first in these agroecological conditions in Serbia on Mollisol. The results contribute to the development of organic farming technology in Serbia. To obtain more long-term data on carbon sequestration, the study will continue with the inclusion of a wider range of variables and factors.

## Figures and Tables

**Figure 1 plants-13-01020-f001:**
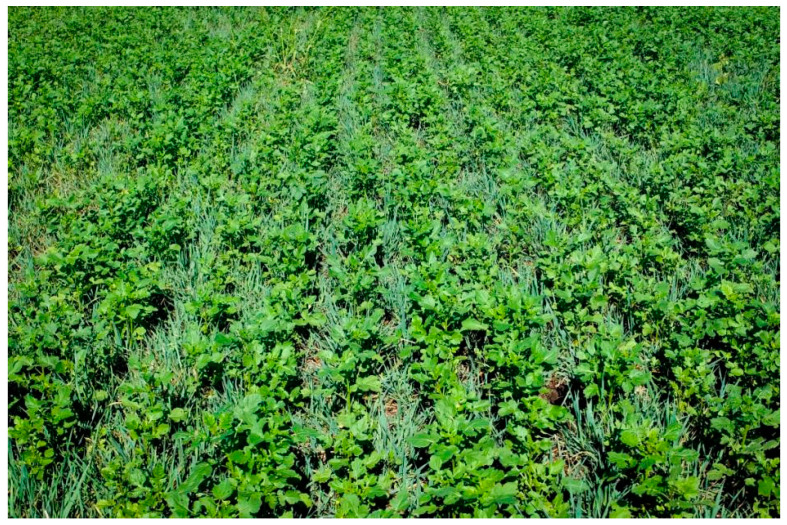
Established mixture of cover crops of self-grown oats and sown white mustard (photo: V. Ugrenović 27 August 2019).

**Figure 2 plants-13-01020-f002:**
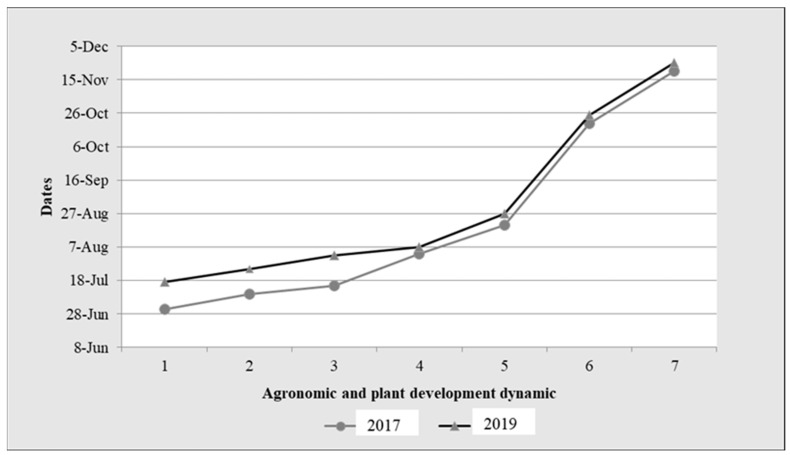
The dynamics of the establishment of cover crops of oats and white mustard in 2017 and 2019. Note: 1. Stubble destroyed; 2. emergence of >50% of oat plants (BBCH 09); 3. sowing white mustard; 4. emergence of >50% of white mustard plants (BBCH 09); 5. land cover establishment (80%); 6. the beginning of flowering of white mustard (BBCH 60); 7. cover crop destruction (BBCH 63).

**Figure 3 plants-13-01020-f003:**
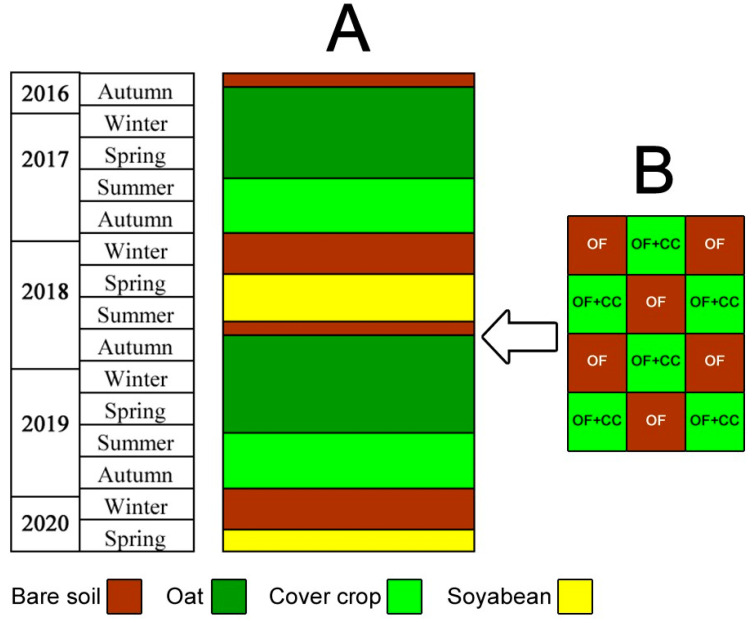
The scheme of the crop rotation (**A**) and experimental plots (**B**) with cover crop (OF + CC) and the control (OF).

**Figure 4 plants-13-01020-f004:**
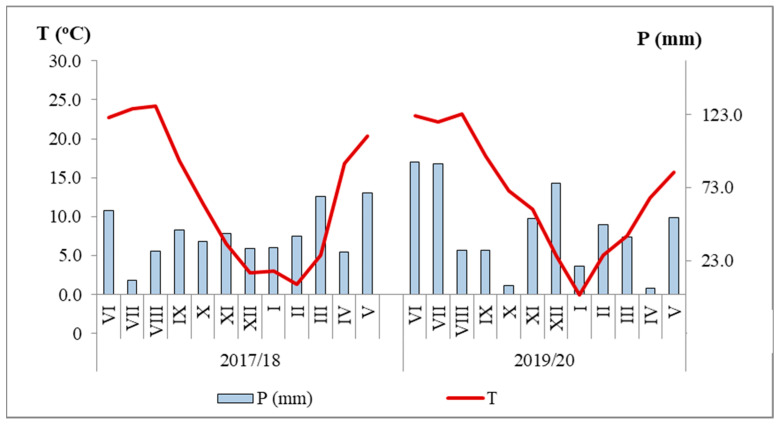
Precipitation (mm) and average air temperature (°C) during the microtrials, Pančevo. The roman numbers are the observation months.

**Table 1 plants-13-01020-t001:** Main soil properties.

Year/Treatment	pH		CaCO_3_	Organic C	Total N	Available	
	nKCl	H_2_O	%	%	%	P_2_O_5_ mg/100 g	K_2_O mg/100 g
2016—initial state	7.54 ± 0.27 ^a^	8.32 ± 0.12 ^a^	12.13 ± 0.07 ^a^	3.26 ± 0.11 ^a^	0.22 ± 0.01 ^a^	19.32 ± 0.14 ^a^	16.40 ± 0.24 ^a^
OF + CC	7.42 ± 0.25 ^a^	8.14 ± 0.10 ^a^	12.12 ± 0.06 ^a^	3.40 ± 0.20 ^a^	0.24 ± 0.01 ^a^	19.44 ± 0.13 ^a^	16.35 ± 0.26 ^a^
OF	7.47 ± 0.30 ^a^	8.28 ± 0.14 ^a^	12.20 ± 0.01 ^a^	3.22 ± 0.12 ^a^	0.22 ± 0.00 ^a^	19.30 ± 0.02 ^a^	16.23 ± 0.13 ^a^
F	0.257	3.672	3.324	2.494	2.984	3.016	1.028
*p*	0.777	0.050	0.064	0.116	0.081	0.079	0.382

Note: OF + CC—organic farming with cover crops; OF—organic farming without cover crops (control); ^a^—means in the same column are not statistically different (*p* < 0.05).

**Table 2 plants-13-01020-t002:** Biomass yield and C/N ratio of oats and white mustard.

	Biomass						Total C and N of Mixed Biomass	
	White Mustard		Oats		White Mustard + Oats			
Year	t ha^−1^	C/N	t ha^−1^	C/N	t ha^−1^	C/N	C%	N%
2017	7.40 ± 0.03 ^a^	8.79:1	1.60 ± 0.04 ^b^	14.31:1	9.00 ± 0.07 ^a^	9.77:1	38.46 ± 0.37 ^a^	3.94 ± 0.05 ^a^
2019	9.21 ± 0.04 ^b^	8.56:1	1.40 ± 0.03 ^a^	13.24:1	10.61 ± 0.03 ^b^	9.83:1	40.31 ± 0.03 ^b^	4.10 ± 0.02 ^b^
F	5102.345	-	53.157	-	1413.873	-	76.368	25.102
*p*	0.000	-	0.002	-	0.000	-	0.001	0.007

Note: ^a,b^ Means in the same column with different superscripts are statistically different (*p* < 0.05).

**Table 3 plants-13-01020-t003:** Soil respiration, microbial biomass carbon and nitrogen (MBC, MBN) after two years of cover crop cultivation in organic cropping system.

Treatment	μg/g CO_2_-C/week	MBC, μg/g	*q*CO_2_	MBN, μg/g	MBC/MBN
OF + CC	1065.81 ± 103.89 ^b^	235.28 ± 5.93 ^b^	4.53 ± 0.44 ^b^	159.43 ± 5.90 ^b^	1.48 ± 0.08 ^b^
OF	469.33 ± 54.42 ^a^	159.83 ± 4.46 ^a^	2.95 ± 0.41 ^a^	135.95 ± 4.30 ^a^	1.18 ± 0.04 ^a^
F	155.190	619.289	41.220	61.981	63.926
*p*	0.000	0.000	0.000	0.000	0.000

Note: OF + CC—organic farming with cover crops; OF—organic farming without cover crops (control). ^a,b^ Means in the same column with different superscripts are statistically different (*p* < 0.05).

**Table 4 plants-13-01020-t004:** Correlation coefficients of soil microbial indices and main agrochemical parameters.

	MBC	MBN	*q*CO_2_	pH_KCl_	pH_H2O_	CaCO_3_	orgC	TN
CO_2_	0.944 ***	0.916 **	0.989 ***	−0.601	−0.802	−0.926 **	0.729 *	0.745 *
MBC		0.841 *	0.890 **	−0.516	−0.846 *	−0.960 **	0.804 *	0.895 **
MBN			0.932 **	−0.732 *	−0.800	−0.833 *	0.793 *	0.588
*q*CO_2_				−0.629	−0.780 *	−0.888 *	0.679	0.654
pH_KCl_					0.275	0.495	−0.712	−0.128
pH_H2O_						0.901 **	−0.634	−0.839 *
CaCO_3_							−0.773 *	−0.873 *
orgC								0.671

Note: ***—*p* < 0.0001; **—*p* < 0.001; *—*p* < 0.005.

**Table 5 plants-13-01020-t005:** Carbon reserves in soils in organic cropping system with and without cover crops.

Treatment	Cover Crop C%	MBC %	CO_2_-C/week %	SOC t ha^−1^	Cover Crop C, t ha^−1^	MBC t ha^−1^	CO_2_-C, t ha^−1^/week
OF + CC	40.31	0.02353	0.1066	62.2545	788.799	0.427	1.935
OF	-	0.01598	0.0469	58.4430	-	0.290	0.851

Note: OF + CC—organic farming with cover crops; OF—organic farming without cover crops (control); MBC—microbial biomass carbon.

**Table 6 plants-13-01020-t006:** Number of specific groups of microorganisms after two years of cover crop cultivation in organic cropping system.

Treatment	Total Number of Microorganismsi (×10^6^/g)	Number of Fungi (×10^4^/g)	Number of Actinomycetesi (×10^4^/g)	Number of Ammonifiersi (×10^5^/g)	Number of Azotobacter MPN †	Free N-Fixers (×10^5^/g)
OF + CC	35.21 ± 3.17 ^b^	11.67 ± 1.28 ^b^	32.58 ± 4.28 ^b^	116.25 ± 9.99 ^b^	462.50 ± 0.76 ^b^	95.59 ± 3.62 ^b^
OF	10.33 ± 2.26 ^a^	7.23 ± 0.63 ^a^	11.34 ± 0.77 ^a^	31.50 ± 2.02 ^a^	173.75 ± 9.05 ^a^	49.01 ± 1.90 ^a^
F	245.596	58.490	143.413	415.048	6059.769	778.103
*p*	0.000	0.000	0.000	0.000	0.000	0.000

Note: Mean values marked with the same letter within one column do not differ significantly (*p* < 0.05; † MPN—the most likely number from Mec Credy table. ^a,b^ Means in the same column with different superscripts are statistically different (*p* < 0.05).

## Data Availability

The data set is unavailable due to privacy restrictions.

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
