# Peer review of "How Do Mixed Cover Crops (White Mustard + Oats) Contribute to Labile Carbon Pools in an Organic Cropping System in Serbia?"

_plants, 2024, doi:10.3390/plants13071020_

Round 1
Reviewer 1 Report (Previous Reviewer 2)
Comments and Suggestions for Authors Dear Authors,the submitted corrected version of your manuscript has been significantly revised and does not need further changes.
Author Response
Dear Reviewer, Thank you for your kind suggestions. We have improved the manuscript in accordance with all comments.
Reviewer 2 Report (New Reviewer)
Comments and Suggestions for Authors
Please see attached file.

The document would be improved by additional read through for English grammar and also for common English usage (word selection, usage, sentence structure, etc.).
Author Response
Dear Reviewers,
Thank you very much for your keen revision. We agree with your comments and suggestions.
Below are the corrections made and answers.
Answers
Lines 27-30: the sentences are rewritten and improved.
Line 32: Keywords are corrected.
Line 34: The Introduction section is shortened, and English is improved.
Lines 63-65: corrected
Lines 84-86: corrected
Line 87, 89 and 90: corrected
Line 94: the section Results is shortened, and English is improved.
Line 136: The Figure 1 is corrected.
Figure 2, lines 149-152: corrected, the NOTE is adjusted.
Line 153: The information on the roller is added.
Table 2: The table is corrected, and statistics are added.
Table 3: The table is corrected, and statistics are added.
Table 4: The significance ranges are given in the NOTE under the table.
Table 6: The table is corrected, and statistics are added.
Discussions: The section is shortened, and English is improved
Line 256: the sentence has been adjusted to be more precise.
Line 258: the sentence has been adjusted to be more precise.
Line 410: The recommended caution was added.
Line 433-435: The recommended caution was added.
Round 2
Reviewer 2 Report (New Reviewer)
Comments and Suggestions for Authors
Nice job. Still need to check English usage throughout. Please check entire manuscript, not just the example below!
For example Line 59-60:
"... dynamics of carbon transformations highly individual. "
Comments on the Quality of English Language
Still need to check English usage throughout.
For example Line 59-60:
"... dynamics of carbon transformations highly individual. "
Author Response
Dear Reviewer,
Thank you for your keen and responsible review of our manuscript. All your comments were very important and useful.
We have revised the manuscript carefully again. We have checked English grammar and spelling with a native speaker.
All changes and improvements are highlighted in the Track change mode.
Also, we found a few minor typing mistakes. Everything was corrected.
This manuscript is a resubmission of an earlier submission. The following is a list of the peer review reports and author responses from that submission.
Round 1
Reviewer 1 Report
Comments and Suggestions for Authors
General comment
The experimental design used, single plots with interplot replicates, does not allow the statistical procedures that have been used inappropriately or reach relevant conclusions regarding the objectives pursued.
Specific comments
Line 46-49: These assertions and data need a reference.
Line 88: Section 2. Results. The Materials ad Method section should come before.
Lines 89-91: Delete
Lines 97-98: “As expected, after 4 years of experiment there was no statistically significant difference in the main agrochemicals parameters between the treatments and the initial state. … Also, the soil acidity level in OF-CC 100 changed toward a more favourable neutral p”.
However there is not any statistical information in Table 1.
Line 100: Check the text.
Table 3: Statistics (letters) are not understood. The comparison should refer to average values, not to individual results.
Table 4: Correlation coefficients. How correlation coefficients were obtained and interpreted? In my opinion, with the experimental design used, it would be a non-normal data distribution, where the correlation coefficient is meaningless. It would be necessary to see the distribution and correlation of the data graphically.
Line 366: “Two way ANOVA was applied to find the effects”. Please, indicate what effects.
Line 367: What is the meaning of the sentence “proposed by Pearson to gain Fisher’s LSD”.
Figure 3: Harrorwing?
Author Response
Dear Reviewer,
Thank you very much for your valuable comments. We are very sorry that we missed such a gross error with statistical data processing.
General comment
The experimental design used, single plots with interplot replicates, does not allow the statistical procedures that have been used inappropriately or reach relevant conclusions regarding the objectives pursued.
We completely agree with your comments. We replaced Table 3 and showed only the average values of the 4 replicates.
Table 5 has been transformed into a graph.
Specific comments
Line 46-49: These assertions and data need a reference.
Answer: We inserted the reference into right place.
Line 88: Section 2. Results. The Materials and Method section should come before.
Answer: The PLANTS journal template was organized like this so we followed the instructions. However, if the editors suggest it, we will change the order of the sections as you suggested.
Lines 89-91: Delete
Answer: deleted
Lines 97-98: “As expected, after 4 years of experiment there was no statistically significant difference in the main agrochemicals parameters between the treatments and the initial state. … Also, the soil acidity level in OF-CC 100 changed toward a more favourable neutral p”.
However there is not any statistical information in Table 1.
Answer: corrected
Line 100: Check the text.
The sentence was deleted
Table 3: Statistics (letters) are not understood. The comparison should refer to average values, not to individual results.
Answer: Table 3 was corrected
Table 4: Correlation coefficients. How correlation coefficients were obtained and interpreted? In my opinion, with the experimental design used, it would be a non-normal data distribution, where the correlation coefficient is meaningless. It would be necessary to see the distribution and correlation of the data graphically.
Answer: Table 4 with correlation coefficients was transformed into a graph.
Line 366: “Two way ANOVA was applied to find the effects”. Please, indicate what effects.
Answer: the mistakes were corrected
Line 367: What is the meaning of the sentence “proposed by Pearson to gain Fisher’s LSD”.
Answer: the mistake was corrected
Figure 3: Harrorwing?
Answer: the mistake was corrected
Reviewer 2 Report
Comments and Suggestions for Authors Manuscript concerns an important problem related with this how mixed cover crops contribute to increasing soil fertility.The studies were conducted over a multi-year period in the certified organic field of the Tamis Institute, Serbia. The biological soil indicators soil respiration, amount of microbial nitrogen and carbon, number of groups of microorganisms were studied.
Modern research methods were used in the study. Organization into sections was well done and paper structure is clear and arranged according to journal style.Introduction is understandable, clear and comprehensive exposing the reader to the topic. Purpose and objectives are scientifically appropriate. Sufficient literary sources are used in the introductory part as a literature reference on the problem.
The methods used are aqurate and correctly described. So described they allow experiment to be reproduced. Appropriate statistics was used.
The experimental results are thoroughly and logically interpreted in the discussion and presented with clarity. Author (-s) compared the results obtained with the experiments and results of previous studies on the relevant subject. The findings are particularly valuable having in a mind importance of the crop studied.
The proposed peer review manuscript is of interest and it deserves to be published in the journal Plants (MDPI).
Author Response
Dear Reviewer,
thank you for your kind comments.
We have corrected inconsistencies and errors according to the reviewers' comments
Round 2
Reviewer 1 Report
Comments and Suggestions for Authors
General comment
From what is indicated by the authors in lines 298-300, it can be deduced that the tests are carried out in a single plot from which several repeated samples are taken (without indicating anything about their position) either to analyze them separately or to incubate. This design is considered poor in obtaining results to which statistical confidence can be assigned. Results that, on the other hand, only show that the incorporation of plant remains into the soil produces completely expected changes in its composition. The authors do not indicate what is new in their results compared to what is already known. It is not considered that with this poor statistical design of the trial, the study is worthy of publication in a journal of the level of Plants.
Specific comments
Table 1: nKCl ?
Line 155-157, and Figure 3: Correlations. In my opinion you have two similar groups of 4 points. That does not define a correlation, despite a high r value.